# Liposomes Loaded with Everolimus and Coated with Hyaluronic Acid: A Promising Approach for Lung Fibrosis

**DOI:** 10.3390/ijms22147743

**Published:** 2021-07-20

**Authors:** Laura Pandolfi, Alessandro Marengo, Kamila Bohne Japiassu, Vanessa Frangipane, Nicolas Tsapis, Valeria Bincoletto, Veronica Codullo, Sara Bozzini, Monica Morosini, Sara Lettieri, Valentina Vertui, Davide Piloni, Silvia Arpicco, Elias Fattal, Federica Meloni

**Affiliations:** 1Research Laboratory of Lung Diseases, Section of Cell Biology, IRCCS Policlinico San Matteo Foundation, 27100 Pavia, Italy; frangipanevanessa@gmail.com (V.F.); s.bozzini@smatteo.pv.it (S.B.); m.morosini@smatteo.pv.it (M.M.); f.meloni@smatteo.pv.it (F.M.); 2Institut Galien Paris-Saclay, CNRS, Université Paris-Saclay, Châtenay-Malabry, 92296 Paris, France; marengo.alessandro88@gmail.com (A.M.); kamila.bohne-japiassu@universite-paris-saclay.fr (K.B.J.); nicolas.tsapis@universite-paris-saclay.fr (N.T.); elias.fattal@universite-paris-saclay.fr (E.F.); 3Department of Drug Science and Technology, University of Turin, 10125 Turin, Italy; valeria.bincoletto@unito.it (V.B.); silvia.arpicco@unito.it (S.A.); 4Unit of Rheumatology, IRCCS Policlinico San Matteo Foundation, 27100 Pavia, Italy; v.codullo@smatteo.pv.it; 5Pneumology Unit, IRCCS Policlinico San Matteo Foundation, University of Pavia, 27100 Pavia, Italy; sara.lettieri01@universitadipavia.it (S.L.); valentina.vertui@gmail.com (V.V.); davidepiloni@live.it (D.P.)

**Keywords:** liposomes, hyaluronic acid, everolimus, lung diseases

## Abstract

Chronic lung allograft dysfunction (CLAD) and interstitial lung disease associated with collagen tissue diseases (CTD-ILD) are two end-stage lung disorders in which different chronic triggers induce activation of myo-/fibroblasts (LFs). Everolimus, an mTOR inhibitor, can be adopted as a potential strategy for CLAD and CTD-ILD, however it exerts important side effects. This study aims to exploit nanomedicine to reduce everolimus side effects encapsulating it inside liposomes targeted against LFs, expressing a high rate of CD44. PEGylated liposomes were modified with high molecular weight hyaluronic acid and loaded with everolimus (PEG-LIP(ev)-HA400kDa). Liposomes were tested by in vitro experiments using LFs derived from broncholveolar lavage (BAL) of patients affected by CLAD and CTD-ILD, and on alveolar macrophages (AM) and lymphocytes isolated, respectively, from BAL and peripheral blood. PEG-LIP-HA400kDa demonstrated to be specific for LFs, but not for CD44-negative cells, and after loading everolimus, PEG-LIP(ev)-HA400kDa were able to arrest cell cycle arrest and to decrease phospho-mTOR level. PEG-LIP(ev)-HA400kDa showed anti-inflammatory effect on immune cells. This study opens the possibility to use everolimus in lung fibrotic diseases, demonstrating that our lipids-based vehicles can vehicle everolimus inside cells exerting the same drug molecular effect, not only in LFs, but also in immune cells.

## 1. Introduction

Lung transplantation is the most difficult challenge among solid organ transplantations. The median overall survival rate of a lung transplant recipient is about 6 years. The long-term outcome of lung transplantation is mainly limited by the occurrence of chronic lung allograft dysfunction (CLAD) which represents a chronic fibrotic reaction of the graft involving the small airways or the interstitial sub-pleural spaces caused by chronic allospecific and aspecific inflammatory injuries [1,2,3,4]. Another clinical setting in which repeated chronic auto-immune inflammatory injuries ultimately cause a diffuse fibrotic reaction of lung tissue, is the interstitial lung disease associated with collagen tissue diseases (CTD-ILD). CTD includes several diseases where ILD represents the principal cause of death [5]. Thus, these end-stage lung disorders share the pathogenic process in which different chronic auto- or allo-specific triggers (acute cellular rejection episodes, insults induced by tissue-specific antibodies, aspecific inflammation due to bacterial or viral agents, gastroesophageal reflux, or pollutants) induce migration, differentiation, activation, and proliferation of myo-/fibroblasts (LFs) in lung tissue. 

Among common immuno-suppressive/immune-modulating strategies that can be adopted in these conditions, mTOR inhibitors deserve particular attention due to their additional anti-fibrotic properties demonstrated by in vitro experiments [6,7,8,9]. Among mTOR inhibitors, we chose everolimus, a synthetic macrolide derivative known to be anti-proliferative and important immune-modulator, which is already used in the clinic for lung transplanted patients to enhance the activity of immunosuppressants, such as calcineurin inhibitors. Everolimus mechanism of action consists in forming a complex with FK binding protein (FKBP)-12, which binds to mTOR, blocking the PI3K/Akt/mTOR pathway [10,11], essential for several cellular processes such as cellular growth, proliferation, and metabolism [12]. The mTOR pathway is also important in in systemic sclerosis (SSc) pathogenesis because through its signaling in SSc fibroblasts it could mediate the immunoinflammatory process typical of ILD in these patients [13]. Although everolimus is very efficient, its important side effects hamper its use in chronic disorders [14]. For instance, in lung transplanted patients, everolimus administration is associated with severe dyslipidemia, decreased wound healing, bone marrow toxicity, and increased risk of lung toxicity [15], often requiring discontinuation of the therapy. 

To reduce these important side effects and to allow the exploitation of its immunosuppressive, antiproliferative, and antifibrotic activities in CLAD and CTD-ILD affected patients, our suggestion is to exploit the potentialities of nanomedicine to target LFs. Cova et al. (2015) already demonstrated that loading everolimus inside targeted gold nanoparticles is a feasible approach and exerts an efficient decrease of in vitro LFs proliferation [7]. However, it was shown that after only 28 days of intra-tracheal administration of gold nanoparticles to mice, they tend to accumulate into alveolar macrophages (AM), thus suggesting a strong accumulation with a potential toxic effect on these cells when a chronic inhalation treatment might be established [16]. To improve the previously described results and achieve a formulation that is readily applicable in clinics, we aim to load everolimus inside more biodegradable nanovehicles, liposomes, to develop a therapeutic option that can be inhaled by a patient delivering everolimus directly inside the lungs. Liposomes that will be tested in this study are specifically directed against LFs thanks to the surface modification with 400 kDa hyaluronic acid (HA), a physiologic ligand of CD44 [17], a glycoprotein overexpressed by LFs and not by healthy bronchial epithelium. Moreover, liposomes are also covered by poly(ethylene glycol) (PEG) since this latest was shown to improve mucopenetration [18].

In this paper we evaluate the in vitro efficacy of these new 400 kDa HA-modified PEGylated liposomes loaded with everolimus (PEG-LIP(ev)-HA400kDa) in inhibiting LFs proliferation, also studying the effect on two other possible main targets: immune/inflammatory cells (AM and lymphocytes).

## 2. Results

### 2.1. Synthesis and Characterization of HA-DPPE Conjugate

The HA-DPPE conjugate was synthesized by coupling DPPE molecules to the HA backbone through EDC/NHS chemistry. The reaction involves the formation of an amidic bond between the activated carboxylic groups of HA and the amino group of DPPE. The reaction was carried out in a mixture of water/tert butanol (52:48 *v*/*v*) to ensure the complete solubilization of both DPPE and HA. The chemical structure of the HA-DPPE was determined using nuclear magnetic resonance spectroscopy (^1^H-NMR) and Fourier-transform infrared spectroscopy (FTIR) spectroscopy. Observing the ^1^H-NMR spectrum (Figure 1) it is possible to note the presence of two peaks corresponding to the aliphatic chain of the DPPE, respectively at 0.9 ppm (terminal -CH_3_ group) and at 1.2 ppm (-CH_2_- groups). The N-acetyl group and the glycosidic protons of the HA were identified at 1.9 ppm and from 3 to 4 ppm.

The degree of substitution (DS), indicating the number of DPPE molecules grafted to HA, was 7.4%. It was calculated, as reported by Saadat et al. [19], by the ratio between the area of the peak corresponding to the methylene group of DPPE (1.2 ppm, green arrow) and the area of N-acetyl proton (methyl) of HA (1.9 ppm, orange arrow).

The correct grafting of the DPPE to the HA backbone was also confirmed by FTIR spectroscopy (Figure 2); the spectrum of HA-DPPE (red line) shows peaks at 1744, 2850, and 2918, corresponding to the carbonyl and acyl chains of DPPE and a large peak at 3400 (black arrow) corresponding to the hydroxyl groups of HA.

### 2.2. Liposomes Formulation and Characterization

Different amounts of everolimus were tested (2, 3, 4, 5, 6, 7.5 and 10 mg) in a PEGylated formulation to identify the best drug/lipid ratio. The total amount of lipid was fixed at 40 mg and the amount of everolimus was progressively increased. Liposomes containing 2, 3 and 4 mg of everolimus were monodisperse with a mean size below 200 nm. From 5 to 10 mg of everolimus a progressive increase in PdI and size values was observed. The encapsulation efficiency (EE) showed the same trend: until 4 mg of everolimus it was above 80%, while above 5 mg everolimus the encapsulation efficiency decreased progressively until 60% (Figure 3 and Appendix A). Furthermore, liposomes formulations prepared with more than 4 mg were not stable and released everolimus after 24 h under storage conditions. Based on these results, the PEG-LIP (non-targeted control) and PEG-LIP-HA400kDa formulations containing 4 mg of everolimus were further characterized and used for the in vitro tests.

PEG-LIP and PEG-LIP-HA400kDa loaded with 4 mg of everolimus were found to be rather monodisperse, negatively charged with a mean size below 200 nm. The PdI values were comprised between 0.20 and 0.25. The encapsulation efficiency was 3.8-fold higher for PEG-LIP than for PEG-LIP-HA400kDa (85% vs. 22%, respectively). Regarding everolimus release by liposomes, we observed a very small amount of everolimus released during the first 5 h, below 5% of encapsulated everolimus for PEG-Lip(ev) and less than 20% of everolimus encapsulated for PEG-LIP(ev)-HA400kDa (Appendix A).

Analyzing storage condition at 4 °C for both formulations, we observed a good stability for at least 3 weeks without any change in mean size and polydispersity index. Table 1 reports the main characteristics of the liposomal formulations.

### 2.3. Cells Internalization of Liposomes

We incubated LFs derived from CLAD and CTD-ILD patients (CD44-positive cells) and 16HBE (CD44-negative cells) with fluorescent PEG-LIP and PEG-LIP-HA400kDa for 4 h at 37 °C analyzing liposomes internalization by flow cytometry and confocal microscopy. The first important observation is that liposome modification with HA400kDa is essential to increase the specific internalization of liposomes in CD44-expressing cells (LFs) (Figure 4) rather than CD44-negative cells (16HBE) (Appendix A). For instance, comparing the percentage of positive population after liposome treatment, only in the case of PEG-LIP-HA400kDa, we obtained a significant increased internalization in CLAD- and CTD-ILD-derived LFs (CD44 positive) compared to 16HBE (Appendix A), in contrast to PEG-LIP (Appendix A). Interestingly, comparing CLAD and CTD-ILD LFs, PEG-LIP-HA400kDa were more efficiently internalized by CTD-ILD-derived LFs (Figure 4b,f) concerning CLAD-derived cells (Figure 4a,d) even if the rate of CD44 protein expression is similar [20].

### 2.4. Effect of Everolimus-Loaded Liposomes on LFs

After assessing that PEG-LIP-HA400kDa are efficiently internalized by CD44-positive cells, we evaluated the effect of everolimus loaded liposomes on LFs assessing cell proliferation using CFSE dye since the primary effect of everolimus is blocking cell proliferation. We incubated LFs derived from CLAD and CTD-ILD with PEG-LIP(ev), PEG-LIP(ev)-HA400kDa and everolimus alone up to 72 h. Following the results obtained in Figure 4, we observed different effects between CLAD and CTD-ILD LFs. In CLAD LFs we observed a significantly reduced proliferation only after 48 h by all treatments, an effect that is restored at 72 h (Figure 5a). In contrast, in CTD-ILD LFs we have the demonstration that liposomes coating with HA400kDa increased significantly drug-induced inhibition of cell proliferation at 72 h compared to PEG-LIP(ev) and everolimus alone (*p* < 0.001) (Figure 5b).

To understand in which phase of cell cycle LFs were accumulated after drug treatment, we analyzed the cell cycle of CLAD and CTD-ILD LFs. By this assay we did not find any difference between PEG-LIP(ev), PEG-LIP(ev)-HA400kDa, and everolimus alone, however, we assessed that the modulation of cell proliferation observed in Figure 5 is related to the accumulation of cells in G0/G1 phase after 24 and 48 h for CTD-ILD LFs (Figure 6b) and CLAD LFs (Figure 6c), respectively.

Knowing that the molecular target of everolimus is mTOR, whose phosphorylation is inhibited by the drug, we evaluated the level of phospho-m-TOR in CLAD and CTD-ILD LFs after PEG-LIP(ev), PEG-LIP(ev)-HA400kDa and everolimus alone treatment. Figure 7a,c showed that in CLAD LFs all three conditions exhibited the same effect without any difference. Regarding CTD-ILD LFs, PEG-LIP(ev)-HA400kDa decreased phosphorylation level of mTOR more than PEG-LIP(ev) and everolimus alone (Figure 7b,c).

### 2.5. Effect of Everolimus-Loaded Liposomes on Immune Cells

Given the known role of everolimus as an immune suppressor, we decided to evaluate the effect of liposome formulation on immune cells. Firstly, we assessed the interaction of fluorescent liposomes on AM and CD3^+^ lymphocytes demonstrating that the modification of liposomes with HA400kDa significantly increased the internalization of liposomes only in the case of AM (Figure 8b,c) compared to PEG-LIP (Figure 8a,b). On the contrary, for lymphocytes we did not see any advantages in modifying the surface of liposomes with HA400kDa (Figure 8d).

Given these observations, we aimed to understand the effect on immune cell activity after treatment with everolimus-loaded liposomes by measuring the secretion of different cytokines.

Regarding AM, we assessed the effect of treatments on IL8 and TGF-β release by AM isolated from BAL of patients affected by CLAD and CTD-ILD. Despite the higher internalization rate of HA400kDa (Figure 8), we did not see any significant effect in the release of IL8 and TGF-β with all treatments (Figure 9). The only modulation that we observed regards CTD-ILD, where PEG-LIP(ev), PEG-LIP(ev)-HA400kDa, and everolimus alone decreased significantly IL8 (Figure 9b) compared to control cells.

Regarding lymphocytes, we focused our attention on CD3^+^ population isolating them from peripheral blood of CLAD and CTD-ILD patients. After incubating CD3^+^ lymphocytes with PEG-LIP(ev), PEG-LIP(ev)-HA400kDa and everolimus alone we assessed the release of IFN-γ (Figure 10a,b) after 24 h, and IL17a (Figure 10c,d) after 48 h of treatment. Results obtained showed that all treatments can significantly reduce the release of IL17a by CD3^+^ lymphocytes of CLAD affected patients compared to control cells (Figure 10c), while there is no modulation of IFN-γ (Figure 10a) in CLAD-derived cells. In contrast, in CTD-ILD CD3^+^ lymphocytes we observed that the release of both cytokines is significantly affected in all treatment conditions (Figure 10b,d) compared to control cells.

## 3. Discussion

In this paper, we demonstrated that the encapsulation of everolimus into targeted liposomes is feasible and can be considered as a new therapeutic option for patients subjected to lung fibrotic disorders consequent to repeated inflammatory/immune-driven injuries. Usually, HA-modified liposomes are developed for cancer treatments [21,22], given the high expression rate of CD44 on cancer cells. However, the results obtained here together with those published previously [20] highlight the possibility to apply this targeting strategy to other clinical settings.

In this paper, we focused our attention on patients subjected to transplant and developing CLAD, which significantly affects long-term survival, and on patients with CTD-ILD, a condition frequently associated with a progressive course towards end-stage lung fibrosis. Unfortunately, no definitive cure exists for both conditions, and treatment at best slow down disease progression in a small percentage of patients and not always with an impact on mortality and quality of life. Of course, the best administration route for these pathologies is the aerosol administration, however, not all drugs can be administered by this route given their potential lung tissue toxicity in a view of a chronic treatment regimen.

For this reason, we decided to exploit liposomes to encapsulate drugs potentially useful for CLAD and CTD-ILD affected patients to reduce side effects increasing their efficacy only in specific cell population. In particular, we modified liposomes surface with HA, given that our target are LFs, the key elements of fibrotic lesions, and considering that increasing HA molecular weight increased its targeting efficiency for CD44, we decided to modify liposomes with a high MW HA: 400 kDa [20,23,24]. Since our final aim is to administer these nanoparticles through aerosol administration, we must also consider the presence of mucus. So, we combined the property of targeting CD44 with mucus-penetration, coating liposomes with PEG, considered as the best strategy for nanotechnologies to overcome the mucus layer [18]. Up to now, all FDA-approved PEGylated liposomes formulation for inhalation are all dedicated for lung infections delivering antimicrobials [25]. Here, we want to show the feasibility to use liposomes formulation also for other drugs.

In this paper, we decided to load inside HA-modified PEGylated liposomes everolimus, an immunemodulator already used for CLAD affected patients, but that induces many side effects [15]. Our analysis demonstrated that PEG-LIP-HA400kDa were rather monodisperse although the polydispersity index was higher than other types of liposomes described in the literature by Iwase et al. These authors reported lower PdI values around 0.18 for PEGylated liposomes containing everolimus having similar composition prepared with ethanol injection [26]; this difference is due to the sonication process performed by the authors after liposome formation. Instead, Pal et al. reported PdI values above 0.2 for liposomes containing everolimus prepared without sonication [27]. PEG-LIP-HA400kDa showed lower zeta potential compared to PEG-LIP because of the presence of the dissociated carboxyl group of HA. The size of PEG-LIP-HA400kDa was larger due to the presence of the HA polymer on the liposome surface. The increase of liposome size, after the addition of HA-phospholipid conjugate, was reported also by Arpicco et al. [23] and Nascimento et al. [28]. This is probably due to the bulky structure of the HA-DPPE that can limit the insertion of everolimus in the liposomal bilayer. Moreover, the ultracentrifugation process, used to eliminate non-associated HA-DPPE, could further induce loss of part of the encapsulated everolimus.

After characterizing HA-modified liposomes, we demonstrated that PEG-LIP-HA400kDa can be efficiently internalized by LFs derived from patients subjected to transplant and developing CLAD and from patients affected by CTD-ILD (Figure 4) and not by CD44-negative epithelial cells (Appendix A). This is crucial to ensure that the drug delivery will be specific only for the pathogenic effectors sparing normal epithelium which usually lacks CD44 expression. Interestingly, we observed different liposome uptake behavior by cells derived from CLAD and CTD-ILD patients, showing a higher internalization rate for CTD-ILD compared to CLAD (Figure 4). This higher uptake is reflected by a significantly higher in vitro activity of PEG-LIP(ev)-HA400kDa on LFs derived from CTD-ILD concerning CLAD. Cell proliferation of LFs derived from CTD-ILD was significantly affected by PEG-LIP(ev)-HA400kDa (Figure 5b). Since the mean level of CD44 expression by LFs derived from CLAD and CTD-ILD resulted analogous, as already reported in a previously published paper [20], we can hypothesize a variable expression of CD44 variants or by a different state of activation of CD44 by LFs derived from CTD-ILD and CLAD. In fact, for cancer cells, it has been reported that even if different cancer cells expressed the same CD44 rate, HA internalization can be modulated by CD44 activation state (which can be affected by external stimuli, post-translational modifications, variant expression, receptor clustering in lipid rafts) [29]. This issue will be analyzed in the future by a specifically designed study. We demonstrated that everolimus-loaded liposomes can deliver a consistent amount of drug into target cells maintaining the molecular activity of the drug. Everolimus delivered by liposomes was able to induce cell cycle arrest in G1/G0 phase (Figure 6) and decreasing phospho-mTOR level (Figure 7), as well as everolimus alone confirming literature data [10,11,30].

These results are encouraging because in chronic fibrotic disorders one of the most important challenges is to block the activation/proliferation of fibroblasts. However, even if the principal effectors are known (LFs), there is a lack of principal molecular regulators. Here, we demonstrated that the modulation of mTOR delivered by targeted liposomes could be an interesting opportunity to inhibit LFs proliferation, confirming the results published previously with other type of nanoparticles [7].

Knowing that everolimus is, firstly, an immunomodulator, we determined the effect of PEG-LIP(ev) and PEG-LIP(ev)-HA400kDa on AM and CD3^+^ lymphocytes derived from BAL and peripheral blood, respectively, of CLAD and CTD-ILD patients. We demonstrated that CLAD liposomes loaded with everolimus did not modulate the release of IL8 and TGF-β (Figure 9a,c). Interestingly, they can decrease significantly IL17a (Figure 10c), a cytokine involved in lung rejection [31], as well as everolimus alone. Regarding CTD-ILD, our liposomes with or without HA decreased significantly IL8 release from AM (Figure 9b) and IFN-γ and IL17a from lymphocytes (Figure 10b,d), as well as everolimus alone.

In conclusion, we can affirm that although we have not always seen an advantage in modifying liposomes with HA, however, we must consider that we are in an in vitro context where the functionalization of nanoparticles is not always evident. Moreover, it is important to highlight the fact that PEG-LIP-HA400kDa demonstrated to be internalized only by CD44 expressing cells and not by CD44-negative ones (16-HBE). Another important point is that our primary aim is to open the possibility to use everolimus in lung fibrotic diseases avoiding side effects, and here we demonstrated that our lipids-based vehicles can vehicle inside cells everolimus exerting the same drug molecular effect, not only in LFs, but also in immune cells. Last, but not least, with all our results we demonstrated that our nano-based therapy can be useful for all lung fibrotic disorders composed of inflammatory and fibrotic phases, having an anti-inflammatory effect together with the anti-proliferative effect on LFs.

The last point that we want to point out is that lipidic-based nanovehicles have been demonstrated to be a very powerful and efficient delivery system of genetic material, such as iRNA or mRNA [32,33]. Regarding pulmonary field the interest in this type of therapy is rapidly expanding, thanks to bioinformatic techniques (i.e., gene expression profiling or biomarkers panels) that allow a more-depth knowledge about the key elements of pathophysiology of diseases. Regarding CLAD up to now researchers and clinicians are trying to identify the best gene/biomarkers [4,34,35]. While the gene therapy approach for chronic progressive fibrosis is under study trying to enhance or silence gene expression using different vectors, including nanoparticles for siRNA delivery [36]. Since several RNA therapies delivered by nanoparticles seem to very promising, we could think about the combination of pharmacological treatment with RNA therapy exerting our synthesized liposomes for CLAD and CTD-ILD affected patients.

Of course, the next necessary step will be the study of PEG-LIP(ev) and PEG-LIP(ev)-HA400kDa on murine models of CLAD and CTD-ILD in order to have clarification about: 1- the real benefit in modifying liposomes surface with HA by in vivo experiments; 2- the efficacy of delivering everolimus locally into the lungs by liposomes, analyzing not only the effect on fibrotic lesions formation, but also on long term effect on mice; 3- the distribution of liposomes after inhalation.

## 4. Materials and Methods

### 4.1. Materials and Instruments

1,2-dipalmitoyl-*sn*-glycero-3-phosphocholine (DPPC), distearoyl-*sn*-glycero-3-phosphoethanolamine-*N*-[methoxy(polyethylene glycol)-2000] (ammonium salt) (mPEG-DSPE) and 1,2-dipalmitoyl-*sn*-glycero-3-phosphoethanolamine (DPPE) were purchased from Lipoid. L-α-phosphatidylethanolamine-*N*-(lissamine rhodamine B sulfonyl) (Ammonium Salt) (Egg LissRhod PE) was purchased from Avanti Polar lipids (Alabaster, AL, USA). Cholesterol (Chol), ethyl(dimethylaminopropyl) carbodiimide (EDC) and N-Hydroxysuccinimide (NHS) were obtained from Sigma Aldrich (Saint-Quentin-Fallavier, France). Hyaluronic acid (HA) 400kDa was purchased from Contipro (Dolní Dobrouč, Czechia) and everolimus from Alsachim (Illkirch Graffenstaden, France). Water was purified using a MilliQ^®^ Reference system from Merck-Millipore. Solvents were of HPLC analytical grade and were provided by Carlo Erba (Milan, Italy).

### 4.2. Synthesis of Hyaluronic Acid-Phospholipid Conjugate (HA-DPPE)

HA-DPPE conjugate was synthesized as described by Saadat et al. with minor modifications [19]. Briefly, 100 mg of HA 400 kDa were dissolved in 20 mL of MilliQ^®^ water and stirred until complete solubilization. Then, 2 M excess of EDC and NHS were added to the aqueous solution to activate the carboxylic groups of HA. The solution was stirred for 2 h at room temperature. DPPE (5 M excess of HA) was dissolved in 20 mL of tert-butanol/MilliQ^®^ water (9:1 *v*/*v*) in the presence of 0.1 mol triethylamine and mixed at 55 °C. The DPPE solution was then added dropwise to HA solution and the resulting mixture was stirred at 60 °C for 6 h followed by stirring at room temperature for additional 18 h. The mixture was dialyzed against MilliQ^®^ water using 3.5 kDa Spectra/Por dialysis bag for 48 h and then centrifuged twice at 1693 g for 30 min to completely remove, by precipitation, the free DPPE. Finally, the aqueous solution was lyophilized to get the solid HA-DPPE product. The reaction was monitored by thin layer chromatography (TLC) using F254 silica gel pre-coated sheets (Sigma-Aldrich, Saint-Quentin-Fallavier, France) and a mixture of chloroform/methanol 70:30 *v*/*v* as mobile phase. TLC was visualized using molybdenum blue solution.

### 4.3. Characterization of HA-DPPE Conjugate

^1^H-NMR spectra of HA, HA-DPPE, and DPPE were recorded using a Bruker Avance 3 HD 400 spectrometer at 400 MHz. HA and HA-DPPE were dissolved in deuterated water while DPPE in deuterated chloroform. FTIR spectra of HA, HA-DPPE, and DPPE were recorded using FTIR Spectrum Two (PerkinElmer, Waltham, MA, USA).

### 4.4. Preparation of Liposomes

PEGylated liposomes containing everolimus (PEG-LIP(ev)) were prepared by ethanol injection method [37]. Briefly, DPPC, Chol, mPEG-DSPE in molar ratio 65:30:5 (40 mg) and different amounts of everolimus (2, 3, 4, 5, 6, 7.5, 10 mg) were solubilized in 1 mL of absolute ethanol and stirred for 10 min at 43 °C. The organic phase was then injected, through an automatic injector (Harvard Apparatus, Pump 11 Elite series) at an injection rate of 1.3 mL min^−1^, in 10 mL of MilliQ water under stirring at 900 rpm. The resulting suspension was stirred for 15 min and then the ethanol and part of water were eliminated under vacuum (20 mbar) at room temperature using a rotary evaporator. The final volume of the liposomal suspension was adjusted to 10 mL with MilliQ water and the liposomes were centrifuged at 1693× *g* for 30 min at 4 °C to eliminate non-encapsulated everolimus that precipitated in the pellet. Hyaluronic acid/PEGylated liposomes containing everolimus (PEG-LIP(ev)-HA400KDa) were prepared by solvent injection method by replacing part of the mPEG-DSPE with HA-DPPE conjugates. The amount of mPEG-DSPE was reduced from 5% to 2.5% molar ratio and 4 mg of HA-DPPE conjugate, previously synthesized, were added in the liposomal formulation. All the lipids (40 mg) and the everolimus (4 mg) were solubilized in 2 mL of tert-butanol/water mixture (60:40 *v*/*v*) and stirred for 10 min at 43 °C. The resulting organic solution was then injected into 10 mL of MilliQ water using an automatic injector equipped with a preheated syringe (injection rate 1.3 mL min^−1^). The suspension was stirred for 15 min at 900 rpm and then the tert-butanol and part of water were eliminated under vacuum (20 mbar) at room temperature using a rotary evaporator. The resulting liposomes were ultra centrifuged at 72,446× *g* at 4 °C for 4 h to eliminate HA-DPPE not associated with liposomes and then the volume of the liposomal suspension was adjusted to 10 mL with MilliQ water. Finally, the liposomes were centrifuged at 1693× *g* for 30 min to eliminate non encapsulated everolimus that precipitated in the pellet. For cellular internalization studies, fluorescent liposomes were prepared by adding in the liposomal formulations 1% molar ratio of Egg LissRhod PE and reducing the amount of DPPC from 65% to 64%. After preparation, liposomes were stored at 4 °C before use and everolimus encapsulation stability was monitored by checking if crystals were appearing since this hydrophobic drug precipitates when not encapsulated.

### 4.5. Liposomes Characterization

The mean size of liposomes and their zeta potential were evaluated by dynamic light scattering using Nano ZS from Malvern (173° scattering angle) at a temperature of 20 °C. Measurements were performed by diluting liposomes by a factor of 10 in MilliQ water for size evaluation and 10 times in NaCl 1 mM for the evaluation of the surface charge/zeta potential. The amount of everolimus encapsulated into liposomes was evaluated by UV spectroscopy using a Lambda 25 UV/VIS spectrometer (Perkin Elmer, Waltham, MA, USA) at room temperature. Liposomes were diluted by a factor 50 in acetonitrile, vortexed and then filtered on 200 nm PTFE filters to remove the phospholipids. The resulting clear solution was analyzed in the range of 400 to 200 nm and the absorption peak of everolimus was identified at 278 nm. The calibration curve was determined by preparing a solution of 20 µg mL^−1^ of everolimus in acetonitrile and subsequent dilution in the range of 20 µg mL^−1^ to 0.5 µg mL^−1^.

Everolimus release by liposomes was assessed at 37 °C at time points of 0, 1, 3 and 5 h. For each time point, 500 μL of freshly prepared liposomes (PEG-LIP(ev) or PEG-LIP(ev)-HA400kDa) were transferred into a vial and diluted 1:5 in PBS (pH 7.4). Then, for time points of 1, 3 and 5 h liposomes were incubated at 37 °C. After the incubation, liposomes were centrifuged at 1693× *g* for 30 min at 4 °C to eliminate non encapsulated everolimus that precipitated in the pellet. The supernatant, containing the liposomes, was collected and the amount of everolimus encapsulated into liposomes was evaluated by UV spectroscopy. For the UV evaluation liposomes were diluted 1:10 in acetonitrile and then processed as described above. Experiments were performed in triplicate.

The percentage of released everolimus, at each time point, was obtained using the following formula:$$\%\;\mathrm{everolimus}\;\mathrm{released}\;\mathrm{at}\;\mathrm{time}\;x=100-[\frac{\left(\mathrm{conc}\;\mathrm{everolimus}\;\mathrm{at}\;\mathrm{time}\;x\right)}{\left(\mathrm{conc}\;\mathrm{everolimus}\;\mathrm{at}\;\mathrm{time}\;0\right)}\;\times \;100]$$

For fluorescent liposomes, the amount of encapsulated Rhodamine was quantified using Vis spectroscopy by diluting liposomes by a factor 10 in ethanol. The resulting solution was analyzed from 650 to 450 nm and the absorption peak of rhodamine was identified at 560 nm. The calibration curve was obtained by diluting an ethanolic solution of 10 µg mL^−1^ of Egg LissRhod PE in the range of 10 µg mL^−1^ to 1 µg mL^−1^.

### 4.6. Cell Culture and Leucocytes Isolation

LFs were isolated from BAL of patients affected by CLAD (n = 3) and CTD-ILD (n = 3) all with chronic progressive fibrosis obtained as previously reported [7]. Briefly, BAL was centrifuge at 400× *g* 10 min, pellet of cells was washed with PBS and counted for cell culture. After 1–3 days of culture, LFs foci started to proliferate. To isolate LFs, foci were harvested to continue their cultivation in DMEM high glucose supplemented with 10% of fetal bovine serum (FBS), 1% of penicillin/streptomycin solution (P/S) and 1% of L-glutamine. The cells isolation from BAL patients was approved by the IRCCS Policlinico San Matteo ethic committee (prot 20100005334) and all patients gave informed consent following the Declaration of Helsinki. After LFs isolation, 6 × 10^6^ cells were cultivated 16-HBE cell line (ATCC) was cultivated in the same medium of LFs.

### 4.7. Liposomes Cell Internalization Analysis

Confocal microscopy. 16HBE, LFs and AM were seeded on 35 mm glass bottom petri dish (Corning Costar, Turin, Italy) at a density of 1.5 × 10^4^ cells. After 24 h, cells were incubated with LIP or LIP-HA400 kDa fluorescently labeled for 4 h at 37 °C. Liposomes were added with the same concentration of Rhodamine. Afterward, cells were washed with PBS, fixed with 4% paraformaldehyde, and DAPI solution was added to label nuclei of cells. Cells were observed by confocal laser microscopy Fluoview FV10i (Olympus, Tokyo, Japan). Flow cytometry. 16HBE, LFs, and monocytes were seeded on 12-well plate at a density of 2.5 × 10^4^ cells. After 24 h, cells were incubated with PEG-LIP or PEG-LIP-HA400kDa fluorescently labeled for 4 h at 37 °C. Liposomes were added with the same concentration of Rhodamine. Afterward, cells were washed with PBS, harvested in cytometer tubes, and analyzed by flow cytometer (Navios, Beckman Coulter, CA, USA) to quantify the fluorescent signal.

### 4.8. Cell Proliferation Analysis

To assess the effect of liposomes loaded with everolimus on LFs derived from CLAD and CTD-ILD patients, we used CellTrace™ CFSE Cell Proliferation Kit for flow cytometry (Thermo Fisher Scientific, Monza, Italy). LFs were seeded on 12-well plate at a density of 3 × 10^5^ cells after labeling them with CFSE dye following the manufacturing instruction. After 24 h, cells were incubated with PEG-LIP(ev), PEG-LIP(ev)-HA400kDa and everolimus alone. In this case, liposomes were added with the same concentration of everolimus (50 nM). After 24, 48 and 72 h cells were harvested and analyzed by flow cytometer to quantify the fluorescent signal of CFSE dye.

### 4.9. Cell Cycle Analyses

LFs were seeded on 12-well plate at a density of 3 × 10^5^ cells and after 24 h were incubated with PEG-LIP(ev), PEG-LIP(ev)-HA400kDa and everolimus alone, with the same concentration of everolimus (50 nM). After 24 and 48 h cells were harvested, washed with PBS and fixed with 70% of cold ethanol. Then, cells were washed twice with PBS and RNase was added to eliminate all RNAs. 50 μg mL^−1^ propidium iodide was added to quantify the amount of DNA inside cells and analyze each cell cycle phase by flow cytometer.

### 4.10. Western Blot

LFs were seeded on 6-well plate at a density of 5 × 10^5^ cells and after 24 h were incubated with PEG-LIP(ev), PEG-LIP(ev)-HA400kDa and everolimus alone, with the same concentration of everolimus (50 nM). After 24 h, cells were washed with PBS, lysed with lysis buffer (50 mM Tris-HCl [pH 7.4], 150 mM NaCl, 10% glycerol, 1% NP-40, protease inhibitor cocktail (Sigma Aldrich) and phosphatase inhibitor (Roche)), gently vortexed for 20 min at 4 °C and centrifuged for 15 min at 13,200 rpm at 4 °C. Supernatants were quantified by Pierce™ BCA Protein Assay Kit (Thermo Fisher Scientific). Twenty micrograms of proteins were loaded and separated in 8% SDS-PAGE. After electrophoresis, the gels were transferred to PVDF membranes (Millipore), therefore blocked (5% BSA in 0.1% Tween 20 TBS) and incubated with the primary Ab (1:1000 in TBST + 5% BSA; overnight at 4 °C): anti-mTOR (1:1000) (PA1518—abcam), anti-p-mTOR(Ser2448) (1:1000) (PA585736—abcam), and anti-β-Actin (1:5000) (MA1-140—Thermo Fisher Scientific). After wash, the membranes were incubated with the appropriate horseradish-peroxidase conjugated secondary Ab (1:5000 in TBST + 5% BSA; 2 h at room temperature; anti-mouse A4416 and anti-rabbit A0545, Sigma). The immunoreactivity was detected by ECL reagents (BioRad, Segrate, Italy), acquired with the Uvitec alliance mini H9 (Uvitec Ltd, Cambridge, UK).

### 4.11. Immune Cells Analyses

AM were isolated from BAL of patients affected by CLAD (n = 3) or by CTD-ILD (n = 3) with adhesion method. Briefly, BALs were centrifuged, and the pellet were washed with PBS and centrifuged again. Afterwards, cell pellet was counted and seeded on 12-well plate at a density of 0.5 × 10^6^ to allow the adhesion of AM. After 1 h at 37 °C, lymphocytes and other cells in suspension were eliminated and fresh medium was added with relative treatments, PEG-LIP(ev), PEG-LIP(ev)-HA400kDa and everolimus alone, to evaluate after 48 h cytokines modulation. IL6, IL8 and TGF-β levels were assessed by SimpleTest ELISA kits (Abcam, Prodotti Gianni, Milano, Italy) collecting supernatants of treated macrophages. CD3^+^ Lymphocytes were isolated from peripheral blood mononuclear cells (PBMCs) of patients subjected to lung transplantation (n = 3) or affected by CTD-ILD (n = 3) were isolated by gradient centrifugation with Lympholyte^®^ (Cedarlane, VT, Canada) in 50 mL conical tube and centrifuged for 30 min at 500 rcf without brake. The PBMCs layer was carefully transferred in a new 50 mL conical tube and diluted with physiologic solution, centrifuged for 10 min at decreasing the rcf of centrifugation up to 200 rcf. To isolate CD3 subpopulation, we used CD3 MicroBeads (Miltenyi Biotec, Bologna, Italy) following the manufacturing instruction. After isolation, cells were seeded on 96-well plate at a density of 0.15 × 10^6^ for the analyses of IFN-γ and IL17a. Lymphocytes were incubated with PEG-LIP(ev), PEG-LIP(ev)-HA400kDa and everolimus alone, at the same concentration of everolimus (50 nM). IFN-γ was quantified by Human IFN-γ ELISpot^PLUS^ kit (HRP) (Mabtech, Nacka Strand, Sweden) after 24 h of treatment, while IL17a was assayed after 48 h of treatment by IL17a ELISpot^PLUS^ kit (HRP) (Mabtech, Nacka Strand, Sweden).

All cell types were also incubated with PEG-LIP and PEG-LIP-HA400kDa fluorescently labeled to evaluate the interaction between liposomes and AM and lymphocytes. Cell were incubated for 1 h at 37 °C and then AM were analyzed by confocal microscopy, while lymphocytes by flow cytometer.

### 4.12. Statistical Analysis

Statistical differences between untreated cells and cells treated with liposomes were evaluated using one-way ANOVA analysis followed by Dunnett or Tukey post hoc test for multiple comparison. All analyses were carried out with GraphPad Prism 5.0 statistical program. A *p*-value < 0.05 was considered statistically significant.

## Figures and Tables

**Figure 1 ijms-22-07743-f001:**
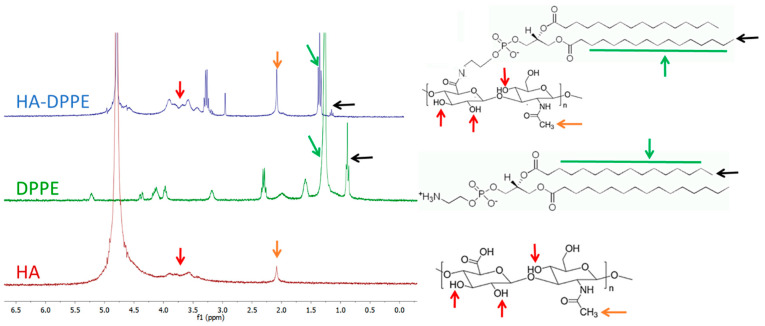
^1^H-NMR of DPPE, HA and HA-DPPE conjugates (left) and corresponding chemical structures (right). Methylene of DPPE group (black arrow). N-acetyl proton (methyl) of HA (orange arrow). Hydroxyl groups of HA (red arrow).

**Figure 2 ijms-22-07743-f002:**
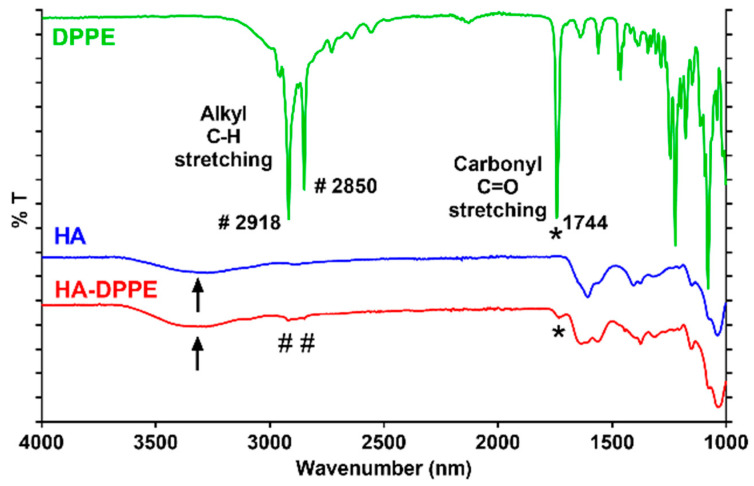
FTIR spectra of DPPE (green line), HA (blue line) and HA-DPPE (red line).

**Figure 3 ijms-22-07743-f003:**
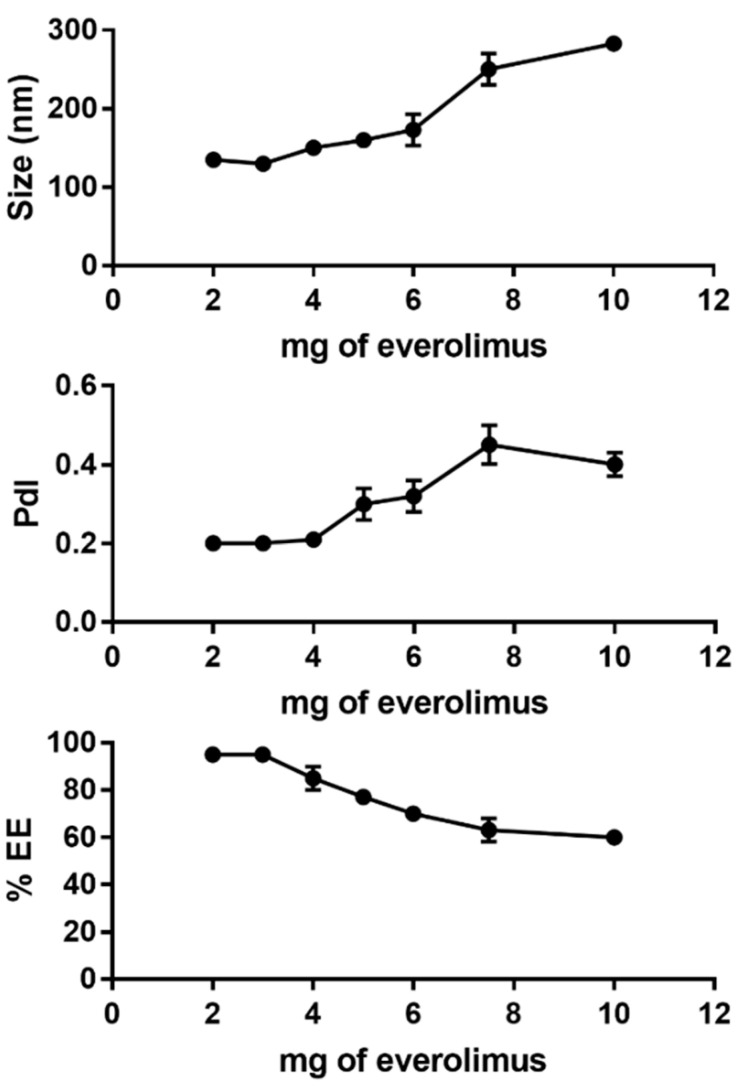
Variations of liposome size, PdI, and EE as a function of mg of everolimus added in the PEGylated liposome formulation.

**Figure 4 ijms-22-07743-f004:**
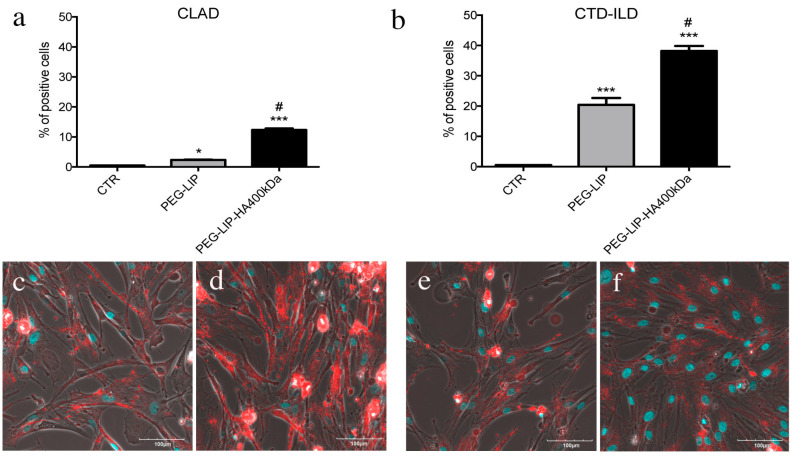
Analysis of fluorescently labeled PEG-LIP and PEG-LIP-HA400kDa internalization in LFs derived from CLAD and ILD patients. (**a**,**b**) Flow cytometry analysis of (**a**) CLAD and (**b**) ILD LFs after 4 h of incubation with liposomes. Data are represented as mean ± SD. *, *p* < 0.05 vs. CTR; ***, *p* < 0.01 vs. CTR; #, *p* < 0.001 vs. LIP. (**c**,**f**) Representative confocal images of LFs CLAD incubated with (**c**) PEG-LIP or (**d**) PEG-LIP-HA400kDa or CTD-ILD LFs incubated with (**e**) PEG-LIP or (**f**) PEG-LIP-HA400kDa. Nuclei of cells = light blue (DAPI); liposomes = red signals. Scale bar = 100 μm.

**Figure 5 ijms-22-07743-f005:**
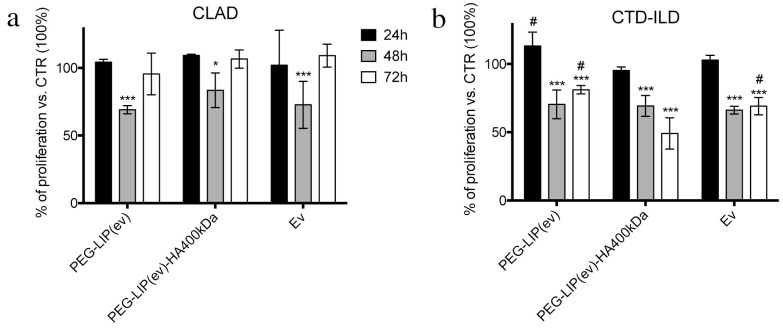
Cell proliferation analysis by flow cytometry of LFs derived from (**a**) CLAD and (**b**) CTD-ILD treated with liposomes loaded with everolimus and everolimus alone (50 nM). Data are represented as mean ± SD of CFSE signals read by flow cytometer for each sample. ***, *p* < 0.001 vs. CTR; *, *p* < 0.05 vs. CTR; #, *p* < 0.001 vs. PEG-LIP(ev)-HA400kDa (One-way ANOVA followed by Dunnett post-hoc).

**Figure 6 ijms-22-07743-f006:**
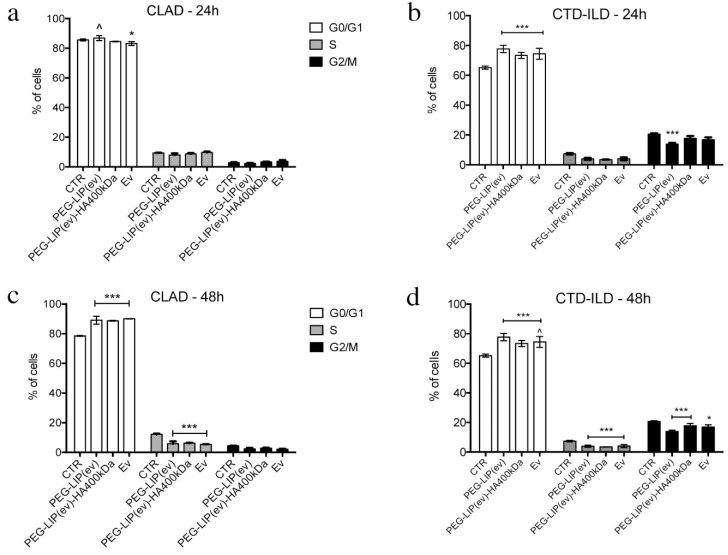
Cell cycle analysis of LFs derived from (**a**,**c**) CLAD and (**b**,**d**) CTD-ILD after incubation with liposomes and everolimus alone (50 nM) for (**a**,**b**) 24 h and (**c**,**d**) 48 h. Data are represented as mean of percentage of cells ± SD in each cell cycle phase. ***, *p* < 0.001 vs. CTR; *, *p* < 0.05 vs. CTR; ^, *p* < 0.05 vs. PEG-LIP(ev)-HA400kDa. (Two-way ANOVA followed by Tukey post-hoc).

**Figure 7 ijms-22-07743-f007:**
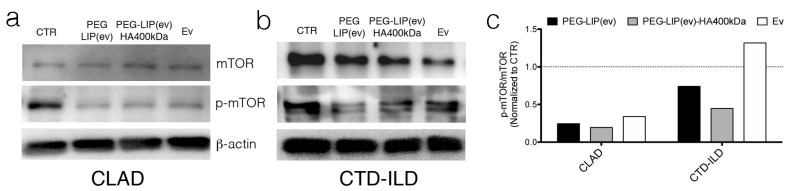
Western blot analysis of mTOR and p-mTOR on LFs derived from (**a**) CLAD and (**b**) CTD-ILD after 24 h of treatment with liposomes and everolimus alone (50 nM). (**a**,**b**) Representative blot of immunedecoration using anti-mTOR, anti-p-mTOR, and β-actin. (**c**) Quantitative analysis of immunoblots representing the expression level of p-mTOR in CLAD and CTD-ILD normalized to CTR = 1.

**Figure 8 ijms-22-07743-f008:**
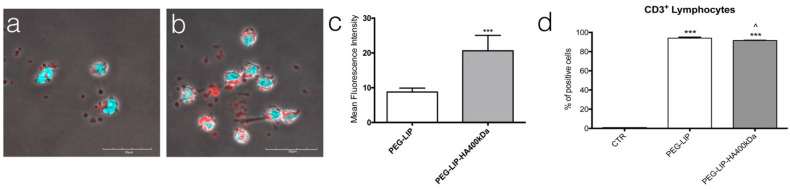
Assessment of liposomes internalization in AM and CD3^+^ lymphocytes derived from CLAD patients. (**a**–**c**) Confocal microscopy analysis of Rhodamine-labeled (**a**) PEG-LIP and (**c**) PEG-LIP-HA400kDa internalized by AM after 1 h of incubation (liposomes = red signal; DAPI = light blue), with (**b**) quantification of mean fluorescence intensity of red signal present in three different ROI in (**a**) and (**c**). Data are represented as mean ± SD, ***, *p* < 0.001 vs. PEG-LIP. (**d**) Quantification of the interaction of Rhodamine-labeled PEG-LIP and PEG-LIP-HA400kDa with CD3^+^ lymphocytes after 1 h of incubation and analyzed by flow cytometry. Data are represented as mean ± SD. ***, *p* < 0.001 vs. CTR. Scale bar = 30 μm. ^, *p* < 0.05 vs. PEG-LIP(ev)-HA400kDa.

**Figure 9 ijms-22-07743-f009:**
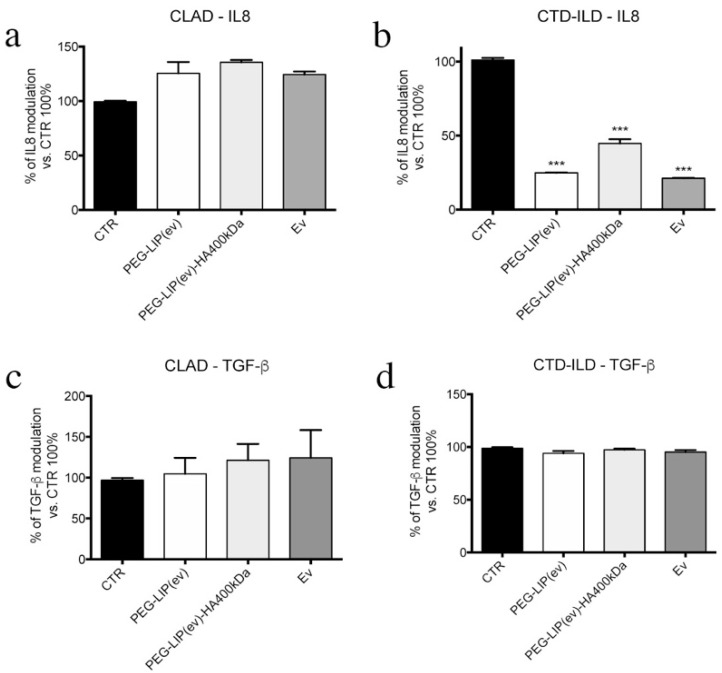
Effect on IL8 and TGF-β release by AM after 48 h of treatments (final concentration of everolimus = 50 nM). (**a**,**b**) IL8 quantification of AM derived from BAL of (**a**) CLAD or (**b**) CTD-ILD. (**c**,**d**) TGF-β quantification of AM derived from BAL of (**c**) CLAD or (**d**) CTD-ILD. Data have been normalized to control cells (100%) and are represented as mean ± SD. ***, *p* < 0.001 vs. CTR.

**Figure 10 ijms-22-07743-f010:**
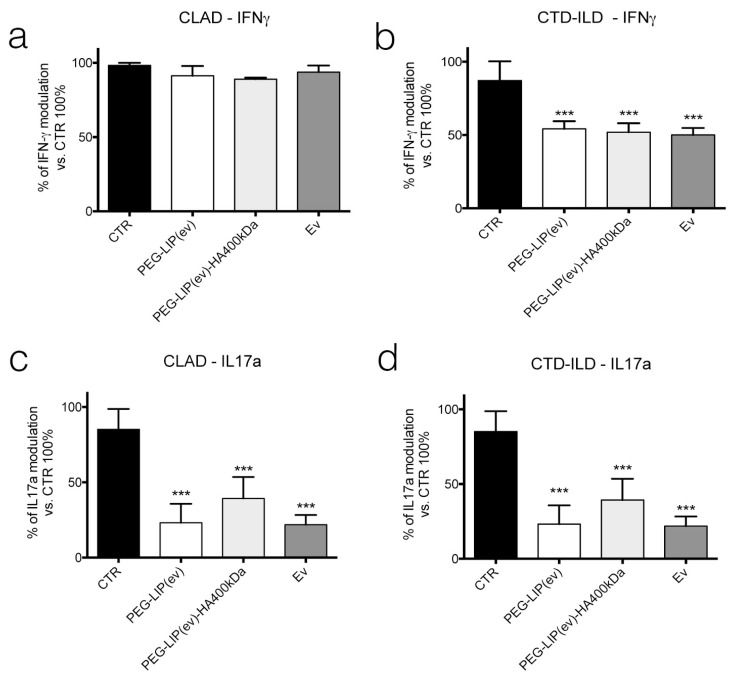
Evaluation of IFN-γ and IL17a release from CD3^+^ lymphocytes derived from peripheral blood of (**a**,**b**) CLAD and (**c**,**d**) CTD-ILD patients after treatment with PEG-LIP(ev), PEG-LIP(ev)-HA400kDa and Ev (50 nM). Data are represented as mean ± SD of number of spots normalized to CTR set at 100%. ***, *p* < 0.001 vs. CTR.

**Table 1 ijms-22-07743-t001:** Characteristics of PEG-LIP and PEG-LIP-HA400kDa loaded with everolimus.

	Size (nm)	PdI	(mV)	EE (%)	Ev μg mL^−1^
PEG-LIP(ev)	152 ± 5	0.21 ± 0.01	−31.7 ± 0.8	85 ± 5	337 ± 26
PEG-LIP(ev)-HA400 kDa	189 ± 15	0.25 ± 0.02	−34.1 ± 0.9	22 ± 3	92 ± 8

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
