# Peer review of "Liposomes Loaded with Everolimus and Coated with Hyaluronic Acid: A Promising Approach for Lung Fibrosis"

_ijms, 2021, doi:10.3390/ijms22147743_

Round 1

Reviewer 1 Report

In this study, Laura et al. reported a novel liposomal treatment approach for lung fibrosis. It is potentially interesting study that suggests their treatment strategy could specifically target fibroblast. However, deeper analysis of mechanistic insights, and clarification of advantage and disadvantage of this treatment are nescessary.

  1. Based on the results presented in Fig 8-10, PEG-LIP(ev)-HA400kDa does not specifically target fibroblasts. Functions of alveolar macrophages, lymphocytes and maybe other cell types would be affected by this treatment. Why the authors can say “PEG-LIP-HA400kDa demonstrated to be specific for LFs”.
  2. Analysis of lung fibroblasts are critical for this paper. What is the definition of LF in this manuscript. Furthermore, isolation methods should be more precisely described, and confirmation of purity should be presented.
  3. The advancement of drug delivery system (DDS) enables application of liposome mediated treatment, and these would impose major impacts on many biomedical areas. The liposome can be used as a drug delivery vehicle not only pharmaceutical drugs, but also siRNAs and mRNA (Nat Nanotechnol 2018;13(3):214-219., Nat Commun 2016;7:11822., Immunity 2020;52(3):542-556 e513.). Discussion and comparison of these novel treatment strategies would draw future directions of DDS based treatments, and clarifies the position of this treatment method. Also, clarification of advantage and disadvantage of this novel treatment presented in this paper are necessary for the next step of this treatment.
  4. In order to use it as a treatment, it is important to verify the effect of systemic administration. The authors should verify in vivo safety, and specificities for lung fibroblasts. Furthermore, PEG-LIP(ev)-HA400kDa could ameliorate fibrosis model? If they cannot perform these experiments, minimally, the authors should explain why they cannot perform in vivo confirmation, and explain the problems that are barriers to overcome in bioadministration.

Author Response

Based on the results presented in Fig 8-10, PEG-LIP(ev)-HA400kDa does not specifically target fibroblasts. Functions of alveolar macrophages, lymphocytes and maybe other cell types would be affected by this treatment. Why the authors can say “PEG-LIP-HA400kDa demonstrated to be specific for LFs”.

When we talk about the specific targeting, we refer to the fact that these targeted nanovehicles are not been internalized by CD44 negative cells, for example 16HBE that are an epithelial cell line. This point is crucial because one of the major problems in lung treatment is the necessity to avoid lung epithelial cells in order to maintain the lung tissue function. Immune cells, in particular alveolar macrophages and lymphocytes express CD44 on their surface (Pandolfi L, Frangipane V, Bocca C, Marengo A, Tarro Genta E, Bozzini S, Morosini M, D'Amato M, Vitulo S, Monti M, Comolli G, Scupoli MT, Fattal E, Arpicco S, Meloni F. Hyaluronic Acid-Decorated Liposomes as Innovative Targeted Delivery System for Lung Fibrotic Cells. Molecules. 2019 Sep 10;24(18):3291. doi: 10.3390/molecules24183291. PMID: 31509965; PMCID: PMC6766933; Schumann J, Stanko K, Schliesser U, Appelt C, Sawitzki B. Differences in CD44 Surface Expression Levels and Function Discriminates IL-17 and IFN-γ Producing Helper T Cells [published correction appears in PLoS One. 2015;10(11):e0143986]. PLoS One. 2015;10(7):e0132479. Published 2015 Jul 14. doi:10.1371/journal.pone.0132479), in fact knowing that we also analyze the impact of our nanovehicles on these cells. Moreover, in this kind of lung fibrotic disorders, the modulation of immune cells could be an added positive activity.

Analysis of lung fibroblasts are critical for this paper. What is the definition of LF in this manuscript. Furthermore, isolation methods should be more precisely described, and confirmation of purity should be presented.

 The isolation method of LFs is described entirely in our previous paper as mentioned in the material and method section (E. Cova et al., “Antibody-engineered nanoparticles selectively inhibit mesenchymal cells isolated from patients with chronic lung allograft dysfunction,” Nanomedicine, vol. 10, no. 1, pp. 9–23, Jan. 2015, doi: 10.2217/nnm.13.208). However, as suggested by reviewer, we add a brief description of LFs isolation from BAL. These cells have been fully described and characterized by our previous paper “Antibody-engineered nanoparticles selectively inhibit mesenchymal cells isolated from patients with chronic lung allograft dysfunction,” Nanomedicine, vol. 10, no. 1, pp. 9–23, Jan. 2015, doi: 10.2217/nnm.13.208.

The advancement of drug delivery system (DDS) enables application of liposome mediated treatment, and these would impose major impacts on many biomedical areas. The liposome can be used as a drug delivery vehicle not only pharmaceutical drugs, but also siRNAs and mRNA (Nat Nanotechnol 2018;13(3):214-219., Nat Commun 2016;7:11822., Immunity 2020;52(3):542-556 e513.). Discussion and comparison of these novel treatment strategies would draw future directions of DDS based treatments, and clarifies the position of this treatment method. Also, clarification of advantage and disadvantage of this novel treatment presented in this paper are necessary for the next step of this treatment.

We thank the reviewer for the suggestion and we added a part in the discussion section

In order to use it as a treatment, it is important to verify the effect of systemic administration. The authors should verify in vivo safety, and specificities for lung fibroblasts. Furthermore, PEG-LIP(ev)-HA400kDa could ameliorate fibrosis model? If they cannot perform these experiments, minimally, the authors should explain why they cannot perform in vivo confirmation, and explain the problems that are barriers to overcome in bioadministration.

We thank the reviewer for his/her suggestion regarding in vivo study. We are doing the in vivo analysis of these nanoparticles in collaboration with two different research groups. Since up to now we have only preliminary results, we prefer to present only in vitro results obtained.

Reviewer 2 Report

The manuscript by Pandolfi and colleagues deals with the preparation of liposomes functionalized with hyaluronic acid (HA) to deliver everolimus to myo-/fibroblasts (LFs). This may result in a new therapeutic option for patients affected by fibrotic lung disorders. The manuscript is well written, and the results are clearly presented. The methodology is robust, and the experiments were carefully done. However, some experiments need to be added.

I have the following observations:

  • Figure 2. Was a statistical analysis done on these data? Please add it.
  • Line 162. No information is reported regarding the storage conditions. Please add it. Also, was the drug loading evaluated during the storage conditions? Was there a partial loss of encapsulated everolimus? If not, this aspect needs to be assessed.
  • The active release from liposomes has not been investigated. Therefore, authors must add this experiment.
  • Table 1. Was a statistical analysis done to conclude that the characteristics of liposomes change after the functionalization with HA?
  • Line 426 and 427: it is not clear what “diluting liposomes 10 times” means. Please revise.

Author Response

Figure 2. Was a statistical analysis done on these data? Please add it.

 The fig 2 is IR spectrum. In our opinion this type of data does not require a statistical analysis. The reviewer was probably misled by the asterisk on the figure. The asterisk in the figure is just a symbol to identify better the IR peaks.

Line 162. No information is reported regarding the storage conditions. Please add it. Also, was the drug loading evaluated during the storage conditions? Was there a partial loss of encapsulated everolimus? If not, this aspect needs to be assessed.

We thank the reviewer for this question. In fact, stability was monitored under storage at 4°C

  • The absence of size and PDI variation during the storage at 4°C.
  • The absence of drug precipitate in the vials during storage conditions. Since everolimus is a highly liphophlic drug, it precipitates in the vial if released.

Some sentences were added in the revised manuscript (lines 470-472)

The active release from liposomes has not been investigated. Therefore, authors must add this experiment.

We did not assess the drug release as liposomes were administered directly in the lung and burst release is not an issue for this route of administration since there is much less dilution than in the blood compartment.

Table 1. Was a statistical analysis done to conclude that the characteristics of liposomes change after the functionalization with HA?

A statistical analysis was not performed but several papers reported similar changes in liposomes characteristics after functionalization with HA:

- Cosco et al. Colloids and Surfaces B: Biointerfaces 158 (2017) 119–126

- Arpicco et al. European Journal of Pharmaceutics and Biopharmaceutics 85 (2013) 373–380

- Hayward et al. Oncotarget 7 (2016) 34158-34171

- Eliaz et al. Cancer Research 61, (2001) 2592–2601,

Line 426 and 427: it is not clear what “diluting liposomes 10 times” means. Please revise.

Diluting liposomes 10 times means diluting the suspension by a factor 10. The sentence was modified in the revised manuscript

Round 2

Reviewer 1 Report

According to authors, they established novel therapeutic approach using everolimus coated liposomes. Overall, more precise description is needed for the enhancement of readability and convincingness.

CD44 is ubiquitously expressed by many cell types, data presented in this manuscript are not enough guarantee the promissingness of this treatment approach. 

Authers assessed HBE bronchial cell lines, lung ”alveolar epihelial cells” and ”endothelial cells” are crucial for lung injury. They must perform additional and comprehensive examination on them.

Also, effects on immune cells should be more precisely examined.

Discussion and comparison of liposome based DDS treatments should be more precisely described, especially other techniques such as  siRNAs and mRNA.  Comparison of different approaches would be important to clarify the  advantage and disadvantage of this novel treatment.

Chronic lung allograft dysfunction and interstitial lung disease associated with collagen tissue diseases are heterogenous diseases, and pathogenesis and treatment strategies are different among each other. Also, importantly, they include NON fibrotic lung diseases. Authors clarify the patients for more detail. Are they diagnosed with chronic progressive fibrosis? 

In vivo applicability is crucial. Authors should explain that the current technique would be directly applicable to in vivo studies ,or this novel strategies are underdevelopment for in vivo use. If so, they should explain problems that are barriers to overcome in bioadministration in vivo. This would be important information for readers.

Author Response

According to authors, they established novel therapeutic approach using everolimus coated liposomes. Overall, more precise description is needed for the enhancement of readability and convincingness.

CD44 is ubiquitously expressed by many cell types, data presented in this manuscript are not enough guarantee the promissingness of this treatment approach. 

Authers assessed HBE bronchial cell lines, lung ”alveolar epihelial cells” and ”endothelial cells” are crucial for lung injury. They must perform additional and comprehensive examination on them.

We already demonstrated in human biopsies the expression level of CD44, showing that normal alveolar epithelia not express CD44. Moreover, in biopsies derived from patient affected by CLAD, CD44 was expressed only by fibroblasts invading bronchiolar lumen and not by other cell types such as endothelial or epithelial one (figure 1 of E. Cova et al., “Antibody-engineered nanoparticles selectively inhibit mesenchymal cells isolated from patients with chronic lung allograft dysfunction,” Nanomedicine, vol. 10, no. 1, pp. 9–23, Jan. 2015, doi: 10.2217/nnm.13.208).

Also, effects on immune cells should be more precisely examined.

Since our first aim is to block proliferation of fibroblasts, responsible of fibrotic lesions. However, knowing the everolimus activity as immunemodulator, we showed only the major cytokines involved in inflammation and rejection (IL17a, IFNgamma, IL8 and TGFbeta), demonstrating that our nanovechiles exert the same everolimus.

Discussion and comparison of liposome based DDS treatments should be more precisely described, especially other techniques such as  siRNAs and mRNA.  Comparison of different approaches would be important to clarify the  advantage and disadvantage of this novel treatment.

As suggested by the reviewer, we introduce a part regarding RNA therapy, but we decided to not go so deep because it is important to note that this is a research paper focused on another approach. We are talking about a pharmacological treatment already used in clinic, for which a discontinuation of therapy is necessary because of the side effects related to this drug. So, our aim is to set the basis to demonstrate the optimal opportunity to use targeted liposomes as drug delivery system for this type of drug, open then the opportunity to test it in mouse model. So, we added in the discussion the argument suggested by the reviewer, but we think it is not appropriate to go much deeper into the very important issue of RNA delivery and its therapeutic possibility.

Chronic lung allograft dysfunction and interstitial lung disease associated with collagen tissue diseases are heterogenous diseases, and pathogenesis and treatment strategies are different among each other. Also, importantly, they include NON fibrotic lung diseases. Authors clarify the patients for more detail. Are they diagnosed with chronic progressive fibrosis? 

As reported in line 487, patients from which we derived LFs are three patients affected by CLAD and three affected by CTD-ILD with a chronic progressive fibrosis. We added, as suggested by the reviewer, a phrase in the material and method section.

In vivo applicability is crucial. Authors should explain that the current technique would be directly applicable to in vivo studies ,or this novel strategies are underdevelopment for in vivo use. If so, they should explain problems that are barriers to overcome in bioadministration in vivo. This would be important information for readers.

Liposomes can be easily nebulized in animals (mice or rats) to conduct preclinical studies (Nebulization of Cyclic Arginine-Glycine-(D)-Aspartic Acid-Peptide Grafted and Drug Encapsulated Liposomes for Inhibition of Acute Lung Injury. Desu HR, Thoma LA, Wood GC. Pharm Res. 2018 Mar 13;35(5):94. doi: 10.1007/s11095-018-2366-9 or Rifampicin-loaded liposomes for the passive targeting to alveolar macrophages: in vitro and in vivo evaluation. Zaru M, Sinico C, De Logu A, Caddeo C, Lai F, Manca ML, Fadda AM. J Liposome Res. 2009;19(1):68-76. doi: 10.1080/08982100802610835) and are even used currently in humans (Amikacin Liposome Inhalation Suspension in Refractory Mycobacterium avium Complex Lung Disease: A Profile of Its Use.  Hoy SM. Clin Drug Investig. 2021 Apr;41(4):405-412. doi: 10.1007/s40261-021-01010-z). Clinical translation of our formulations is therefore very likely provided in vivo experiments are showing efficacy in animals. For animal studies, nebulization devices are available or inhalation chambers can be used with commercial nebulizers. For human studies many different types of nebulizers exist on the market. Once administered, droplets of liposomes suspensions deposit in the lungs and may interact with cells provided they can cross the mucus barrier. This aspect was discussed extensively in the manuscript (lines 305-310). Crossing the mucus is the reason we developed PEGylated formulations of liposomes as many studies in the literature have reported PEG facilitates mucus crossing (REFRENCE 18 of the manuscript)

Reviewer 2 Report

I thank the authors for providing the revised version of their manuscript. Still, I have the following comments:

  • I apologize for my mistake. The comment "Figure 2. Was a statistical analysis done on these data? Please add it." should be referred to Figure 3. Of course, I know what FTIR spectra are and how to interpret them to understand that no statistic is required for that type of data. However, please add statistic in Figure 3. 
  • Still, it is not specified in the text what the storage conditions are. Please add that for storage conditions, you mean 4 °C.
  • The release profiles, in my opinion, are still mandatory. It is not only a matter of the burst release. You demonstrated that a large part of the liposomes is uptaken by cells after 4 h. Moreover, you also specified that no drug was released in 3 weeks, but in storage conditions, and thus at 4 °C. What happens in 4 h at 37 °C? If, hypothetically, all the active is rebased by the liposomes in this short time, it makes no sense to use them to deliver the active into cells. Thus authors must investigate the drug release from liposome in these conditions.

Author Response

I thank the authors for providing the revised version of their manuscript. Still, I have the following comments:

  • I apologize for my mistake. The comment "Figure 2. Was a statistical analysis done on these data? Please add it." should be referred to Figure 3. Of course, I know what FTIR spectra are and how to interpret them to understand that no statistic is required for that type of data. However, please add statistic in Figure 3. 

We understand the concern of the reviewer regarding statistical differences in all data. Regarding size and PDI measurement the data present on both graphs are the results of a size distribution, meaning a repartition of values on a graph from which the mean diameter is extracted. In the literature most of the authors consider that a difference of 10% is a change. Moreover, the most important value is the PDI that in the present case increase starting from 4 mg telling us that important augmentation of particle size distribution has occurred. We hope the reviewer agrees on our rationale which also explains why in articles published in more specialized reviews, people never make statistics on values that are already based on statistical evaluation.

  • Still, it is not specified in the text what the storage conditions are. Please add that for storage conditions, you mean 4 °C.

The storage conditions were added in the first revision line 495: “After preparation, liposomes were stored at 4°C before use and everolimus encapsulation stability was monitored by checking if crystals were appearing since this hydrophobic drug precipitates when not encapsulated.”

  • The release profiles, in my opinion, are still mandatory. It is not only a matter of the burst release. You demonstrated that a large part of the liposomes is uptaken by cells after 4 h. Moreover, you also specified that no drug was released in 3 weeks, but in storage conditions, and thus at 4 °C. What happens in 4 h at 37 °C? If, hypothetically, all the active is rebased by the liposomes in this short time, it makes no sense to use them to deliver the active into cells. Thus, authors must investigate the drug release from liposome in these conditions.

As suggested by the reviewer, we added the everolimus release analysis in results with figure as figure S2 and in material and methods.

Round 3

Reviewer 1 Report

It is promissing strategy, but the advantage and disadvantage of this treatment should be further validated in another study.

Author Response

We are accumulating evidences about the use our synthesised liposomes in two different animal models:

1- CLAD mouse model in collaboration with University of Messina (Italy);

2- CTD-ILD in collaboration with a research group of Instituto de Investigación Sanitaria-Princesa (Madrid).

Reviewer 2 Report

I thank the authors for providing the revised version of their manuscript. I am satisfied with the responses provided for my observations regarding the storage conditions and the release. Regarding the statistical analysis for Figure 3, I can understand the explanation provided by the authors for the data regarding the mean size and PDI. Still, no explanation is provided for the lack of statistical analysis regarding EE.

Author Response

We added as figure S2 the statistical analysis regarding EE.